# Artificial Intelligence: A Blessing or a Curse. Discussing Security Concerns of Diagnostic Models in Radiological Assessment

Anonymous Full Paper
Submission 10

## Abstract

Radiology is increasingly adopting AI-based workflows, which provide promise but also introduce new security concerns. The goal of this research is to enhance the security of these workflows by evaluating the risks of data poisoning attacks using the Fast Gradient Sign Method (FGSM) and Carlini-Wagner (C&W) techniques. Detection methods commonly employed in financial fraud are evaluated to assess their effectiveness in this context. Knowledge distillation is also explored as a defense mechanism against data poisoning, offering a potential mitigation strategy. By conducting these evaluations and proposing defenses, this research aims to contribute to the discussion of more robust deployment of AI systems in real-world radiology applications.

## 1 Introduction

Radiology has had a tendency to be an early adopter of new technologies. Medical imaging is at the forefront of research, since it is part of the initial processes for making a diagnosis [1]. The use of AI, along with the discipline's eagerness, is part of a tradition. There has been the X-ray, MRI, and CT scan, each readily adopted into workflows. AI marks a continuation of that trend as it becomes more commonplace. However, AI fundamentally differs from previous technological advancements. Other imaging technologies simply provide more data, but AI seeks to complement the radiologist. It is also more mercurial by nature. Other technological advancements are deterministic. If one takes an X-ray, there is a high degree of certainty that the image will yield the same result. This is not the case with AI; it is stochastic. Given images, models simply report what is most probable from their perspective. It lacks the same promise of consistency. Software engineers approach this issue by defining key performance indicators and creating thresholds for performance. They ensure that the model will perform above a certain percent accuracy or with a degree of specificity [2]. This measure is usually based on the confusion matrix and Area Under the Curve.

### 1.1 Data Drift

To ensure that performance metrics are maintained, software engineers must regularly update and maintain the models under their stewardship. The main challenge is that the performance of models tends to fluctuate and degrade over time [3]. This phenomenon is known as data drift. For example, consider a model trained to analyze X-rays of legs from people in Scandinavia. If there were a significant migration of shorter individuals into the population, the model's performance would likely decline. The AI relies on identifying certain characteristics in specific locations. Due to the variation in height, discrepancies could arise between what the AI has learned and the data it encounters. Thus, leading to degraded performance. This is data drift: the discrepancy between the environment in which the AI operates and the data on which it was trained. To address these issues, it is common practice to update models with more representative data. While this practice aims to create more robust models, it also opens up opportunities for bad actors to manipulate the results of models for malicious purposes.

### 1.2 Data Poisoning

One technique to keep models aligned with the current deployment context is to train them on data that the AI processes during inference. Essentially, the model may save the input provided by the user along with the conclusion made, thereby reinforcing its predictive ability. However, this opens the door for bad actors to craft payloads designed to intentionally degrade the model, encouraging it to learn incorrect associations [4]. This is known as data poisoning. Consequently, while your model updates to avoid data drift, it could inadvertently poison itself. One critical aspect of data poisoning to understand is that human inspection alone would not detect anything amiss. The alterations made to poison the data are not something the human eye can easily discern, adding a layer of stealth to these attacks if an adversary is determined to undermine.

Data Poisoning is an increasing significant concern through advancements in deployment of AI within clinical settings. For instance, Project MONAI is a framework that has become popular and focuses on clinical use-cases. The MONAI Deploy extension has already been utilized in the wild and implements

continuous learning capabilities [5]. With increased deployment, the risks associated with using AI in clinical workflows will increase with time.

## 1.3 Contribution

There are several aims this paper seeks to target. The first is to investigate adversarial attacks. Two of the most common attacks are FGSM and C&W. The literature suggests that luminosity is a factor in the ability to conduct attacks. [6]. Thus, the data from 2017 RSNA Pediatric Bone Age Challenge is investigated [7]. The contrast between the brightness of the bones and the black background make this analysis within the domain of adversarial attacks of medical images significant. The dataset is also selected because the images are more sparse, whereas typical scans usually contain significantly more noise. By using X-rays, the manipulations become clearer and more understandable, and the structures of the bones are more defined. It is also multi-class. A second dataset is used for only segmentation. It is the Kvasir-SEG dataset [8]. This dataset is designed for the segmentation of polyps. This is valuable as a contrast to Bone Age. The images are color and are of the GI tract; these are visually more complex. By conducting this analysis, it provides guidance how adversarial attacks affect segmentation and not only classification.

The second aim is to better understand the applicability of Benford's Law to medical imaging. There has only been some initial investigation in the literature [9]. This requires consensus building to uncover the strengths and limitations of this approach.

The last aim is to explore how utilizing different loss functions within distillation affects their defensive capability and how different tasks are affected by adversarial attacks.

## 2 Background

Perturbations are the primary attack methods used against computer vision models and pose a significant risk to models deployed in the healthcare space. Therefore, it is essential to understand these attacks and how they are used in context. Additionally, detection methods from other fields, such as Benford's Law, are worth considering. Finally, it is important to explore defenses against these attacks to learn how developers can be both proactive in detection and reactive by fortifying their models if poisoned data slips through.

## 2.1 Computer Vision

YOLO (You Only Look Once) and U-Net are the focus of this discussion as they represent the most widely used models. YOLO is a productionized object detection and classification model that is widely used in industry. The first iteration of YOLO was published in 2016 [10]. There have been upwards of a eleven versions, and it has been foundational in computer vision. YOLO will be primary focus of this study. U-Net is a convolutional neural network (CNN) like YOLO, but it is designed for medical segmentation. It utilizes an encoder to compress the inputs into the feature space and then uses a decoder to up-sample to recover spatial information to create the segmentation map [11]. A standard U-Net is utilized in this work.

YOLO will be the focus of the work as it is generally more popular. YOLO segments an image based on labeled images with annotated bounding boxes. It then predicts the presence of particular classes within the image during inference. YOLO is currently the benchmark for object detection tasks and has been enhanced through a series of innovations. The first key innovation of YOLO is the use of Cross-Stage Partial Networks [12], which minimizes the computational load required for convolutions by employing a partitioning strategy. The second innovation involves Path Aggregation Networks [13], a technique that enables the development of sub-networks for more robust feature pooling. Lastly, YOLO produces three feature maps that are fused to create a more informative output, which is then further interpreted [12]. These innovations have solidified YOLO as the standard in object detection and classification.

## 2.2 Attacks

There are two main attack methods evaluated in this endeavor. The first is Fast Gradient Sign Method. This method is far simpler and is more well suited for general attacks.

$$x' = x + \epsilon \cdot \text{sign}(\nabla_x J(\theta, x, y)) \tag{1}$$

In FGSM, there is your input, x. In this context, it is an image. $\epsilon$ is essentially the amount of manipulation the attack should inflict to the image. It is a tuning parameter. The larger the value, the greater the amount of perpetuation. However, this increases the visibility of attack and increases the likelihood that your attack will be detected. $\nabla_x J(\theta, x, y)$ is the gradient of the loss function and is used by FGSM to identify the directionality of perturbations to maximize the model's prediction error [14]. This can be better conceptualized when considering a prediction landscape and pushing the poisoned data against the gradient.

$$\min_\delta \|\delta\|_p + c \cdot f(x + \delta) \tag{2}$$

Carlini & Wagner is the second attack evaluated. The C&W attack is an optimization-based approach.

The perturbation $\delta$ is measured with an $L_2$ norm in most cases. This helps to constrain the space of the perturbation while still allowing it to misclassify the input. The constant $c$ is a tuning parameter. It helps to shrink $\delta$ to the point where it may still misclassify inputs without it being larger than necessary, which could draw attention and increase the chance of detection [15].

## 2.3 Detection

To detect these attacks, techniques from other domains are being explored. The investigators in [9] apply methods grounded in Benford's Law to detect such attacks. Benford's Law suggests that the leading digit distribution of natural datasets follows a logarithmic function. In their analysis, the goal was to test whether the leading digit distribution of pixel values deviates from that of natural images. The approach involved treating the input image as a vector, computing a gradient, transforming the data, and then comparing the frequency of the first digits using the Kolmogorov-Smirnov test [16]. The gradient magnitude of the input image and the Discrete Cosine Transformation typically adhere to Benford's Law, and these were used in the comparison. The Kolmogorov-Smirnov test was employed to determine the divergence of these distributions. Using these techniques, the investigators successfully identified 94.7% of a Projected Gradient Descent attack with infinity norm and 81.8% with L2 norm.

## 2.4 Defenses

Proactive measures can be taken to mitigate attacks, rather than simply focusing on reducing response times. In [17], the investigators introduce knowledge distillation, a strategy designed to make models less susceptible to adversarial input by facilitating knowledge transfer. Essentially, class probability vectors are fed into a smaller secondary model that produces a more discrete result. While the primary goal is to enhance the model's robustness, knowledge distillation is designed to have minimal impact on the model's architecture, maintain accuracy, and preserve speed. The technique significantly reduced the success rate of adversarial sample crafting from 95.89% to 0.45% on the MNIST dataset [17]. Additionally, the distilled model experienced only a 1.28% drop in accuracy, without requiring significant changes to the base model.

## 3 Literature Review

U-Nets have been developed for medical segmentation and have been a focus of attacks along with YOLO. In [6], the investigators applied FGSM to U-Nets. They found image luminosity is a factor

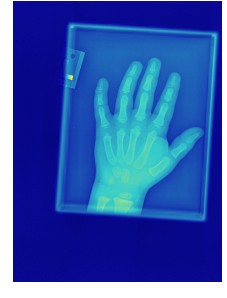 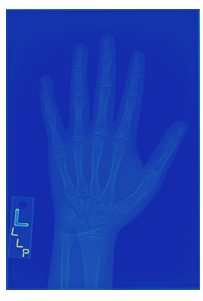

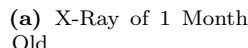

**(a)** X-Ray of 1 Month Old

**(b)** X-Ray of 5 Month Old

**Figure 1.** Comparison of X-Ray Features

for attack success. This suggestion is in part motivated using the Bone Age dataset in our analysis. They used the Dice Score and found a 40% drop after applying only FGSM to four varieties of U-Net. Furthering the investigation of FGSM attacks, the investigators in [18] also applied C&W attacks and attempted to defend against both. However, this work was outside of the medical context. They found that distillation was effective at the mitigation of FGSM but not C&W. The loss function used in their distillation process utilized a singular weighting parameter to balance loss of the student model with the distillation loss, which can be thought as agreement between the student and teacher model.

## 4 Detection Techniques

To guard against these attacks and to ensure quality checks for AI, qualitative and quantitative techniques have arisen. The first technique is more qualitative. It is inspecting saliency plots to understand which aspects of the input are most significant [19]. Applications of Benford's Law are more quantitative measures for detecting manipulation.

When reviewing these two plots, one can interpret which features are most significant to the classification of hand bone ages. For the one-month-old, the structure and definition of the carpal bones appear to be most significant. For instance, one might notice the roundness due to the lack of structure. Additionally, there is increased highlighting in the larger gaps between the phalanges. For the five-month-old, the carpal bones are more well-defined, and there is notable highlighting of the radius and ulna in the scan.

One issue to consider is the presence of the "L" marker. This marker is intended to help radiologists orient the scan by indicating the left side of the person being scanned. The strongest highlighting is found along the contours of the letters on the marker. The concern is that the model might be associating the marker with older hands. Younger hands are smaller and often require zooming in, which can

**(a)** Benford's Law Applied to C&W Attacked Scan From Kvasir

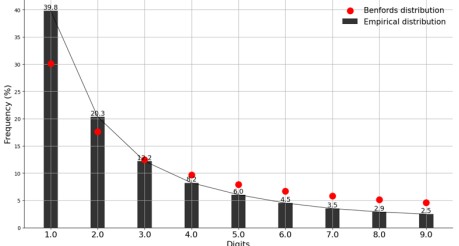

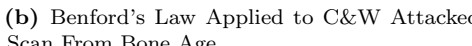

**(b)** Benford's Law Applied to C&W Attacked Scan From Bone Age

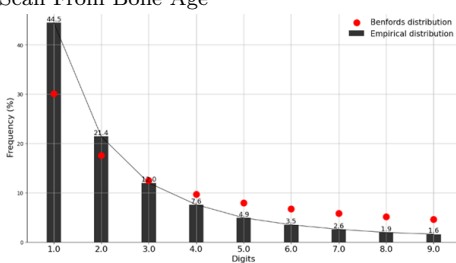

**Figure 2.** Comparison in Digit Frequency

obscure the marker and implicitly train the model to recognize the marker as a feature associated with older hands. If the marker were superimposed on younger hands, they might be incorrectly interpreted as older due to this spurious relationship. Thus, models may sometimes hamper their own learning without external influence and may simply learn problematic features.

## 4.1 Benford's Law

Benford's Law is an observation regarding the frequency of digits. For instance, the number "1" should appear in the first digit of a number roughly 30% of the time. The number "2" should appear as the first digit in a number 18% of the time. This pattern in frequency decreases with the higher the digit. The underlying process causing this distribution is not well understood, but it provides a useful benchmark to suggest if the data has been manipulated not. In [9], a cosine transform is applied to the images which allows Benford's Law to hold.

A scan of a five month old is evaluated to determine if an attack by C&W could be determined by inspection for Bone Age. For Kvasir, C&W is also used on the image of a polyp.

When observing how these distribution differ from the expectation, it is suggestive that the has been manipulated. This is most noticeable when observing the frequency of "1" in both datasets.

A limitation to consider is that when applying the Kolmogorov-Smirnov test, the p-value tends to produce too many false positives. It is likely that this test should be applied to entire datasets rather than individual images. If Benford's Law is to be applied to specific images, it is more useful to inspect the distribution through graphs or to use more advanced versions of Benford's Law to increase the robustness of the detection. For instance, it may be more robust to jointly look at the first and second digits to avoid false positives.

When evaluating Benford's Law on an image attacked by the FGSM, the technique did not perform well. This holds for both datasets. The attack caused the frequencies to more closely align with the expected frequencies, leading to false negatives. Observing entire datasets may help mitigate these deviations in individual scans and determine whether there has been a systematic attack on the dataset.

## 5 Results

YOLO is a model for object detection, and in this formulation, the ages of bones were treated as classes to segment. The accuracy measurement reflects how well YOLO was able to determine a bounding box for the correct age of the bone from the scan. After training YOLO for 10 epochs, the model achieved an accuracy of 64% in segmenting the images to determine the correct class. This is a reasonable baseline, with the highest accuracy being 78% for 1-month-old scans. However, it performed the least well for 8-month-old scans, with an accuracy of only 28%.

When applying the FGSM, the average accuracy drops to 17% on the adversarial example set. For the less representative classes, YOLO was completely fooled. For instance, none of the 8-month-old scans were correctly classified. However, for larger classes, the model performed slightly better, with 5-month-old bones reaching 32% accuracy and 4-month-old bones reaching 26

As for the C&W attack, the overall accuracy drops to 12%. YOLO was again tricked by the less representative classes. For newborns, none of the poisoned test set scans were correctly classified. Accuracy remained close to 0 for 7-month-old and 8-month-old scans. The majority class, 5-month-olds, saw a performance of 35%.

U-Net is a model suited for segmentation. It achieved a mean average precision of 42%. This drops to 22% under FGSM and 19% under C&W. It was also trained for 10 epochs and adhered to the same general process that is used to evaluate YOLO.

## 6 Knowledge Distillation

Knowledge Distillation is a technique designed to guard against adversarial examples. The effectiveness of the previous attacks is due to the inherent fragility of CNNs, where small perturbations can significantly manipulate inference. Knowledge distillation involves a student-teacher model approach. In this instance, the teacher model is YOLO. The confidence score of the predicted class is extracted from YOLO and used as a new input. This score, along with the class (in this case, the age of the infant being scanned), serves as input for a secondary ResNet model. The ResNet is then evaluated both with and without distillation.

For the U-Net, instead of feeding the output directly into the student model, it is added to the loss function. For segmentation tasks, basing the loss on ground truth with the teacher model's soft targets supports student model in learning segmentation boundaries from the teacher. The loss used is

$$\text{loss} = \alpha \cdot \text{label\_loss} + (1 - \alpha) \cdot \text{distillation\_loss} \quad (3)$$

This formulation allows for a more precise weighting to be applied to learning the ground truth and agreement with the teacher model.

To test YOLO, 500 adversarial examples were generated using the C&W attack, and 10,000 adversarial examples were generated using FGSM to create a poisoned version of the original dataset. C&W is too computationally intensive to warrant so many examples, since the adversarial examples are more potent. As for Kvasir, 1,000 examples were created under FGSM with 100 with C&W with the same reasoning.

When reviewing the example model without distillation, its ability to classify images is very limited. The model is essentially only capable of correctly classifying bones from 4-month-olds and 5-month-olds, while misclassifying all other age groups. This suggests that the model is not effectively learning the distinguishing characteristics of the other classes and is instead relying on the prominence of certain classes in making its predictions.

In contrast, when applying distillation, the model demonstrates a greater ability to differentiate between characteristics across classes. This improvement is evident in the confusion matrix, which shows a more diverse distribution of predictions. The distilled model is attempting to generalize and learn from the data. However, the limitations of this model may indicate that the teacher model (YOLO) has not been sufficiently trained. The YOLO model was only trained for 10 epochs, yet there is still a significant difference in performance between a standard ResNet and a ResNet that has undergone distillation. However, the dataset is large and is using

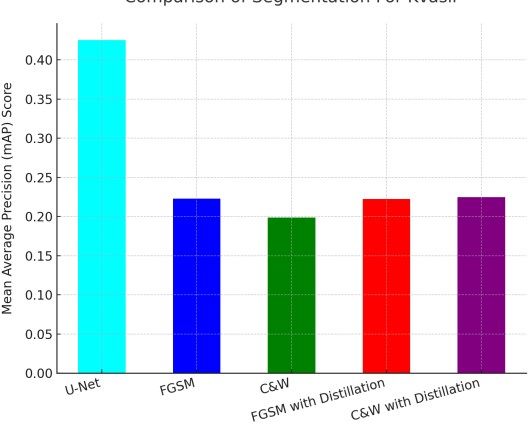

**Figure 3.** Comparison of Attacks Under Distillation for Kvasir

a foundational model to conduct transfer learning, so the concern may be overstated. Also, the difference in accuracy is the finding and not the absolute accuracy.

When evaluating the effectiveness of distillation in mitigating C&W, the results are not particularly promising. Without distillation, the model struggles to generalize effectively. Although the distilled model performs marginally better, such as predicting 7-month-olds more accurately, it largely fails to cope with the attack.

As for the segmentation results, distillation was effective at improving model robustness under C&W. However, it becomes less robust to FGSM and informs a trade-off in how loss functions should be formulated for targeting specific attacks.

## 7 Conclusion

In this study, two attack types were applied to the RSNA Bone Age dataset and Kvasir using FGSM and C&W. FGSM, while simple and effective, is harder to detect, with Benford's Law failing to catch its perturbations. C&W, being more complex, is more damaging but also more noticeable. Knowledge distillation showed some effectiveness, particularly against FGSM when considering the attack on YOLO. However, it is ineffective when considering U-Net and FGSM. This is likely attributable to the loss function that is able to more effectively protect against C&W.

It is recommended to apply Benford's Law for detecting severe attacks, while knowledge distillation can help mitigate simpler ones. Future work should focus on understanding how loss function formulations factor into distillation effectiveness and the robustness of Benford's Law in the medical image domain to further support robustness initiatives.

# A   Appendix A

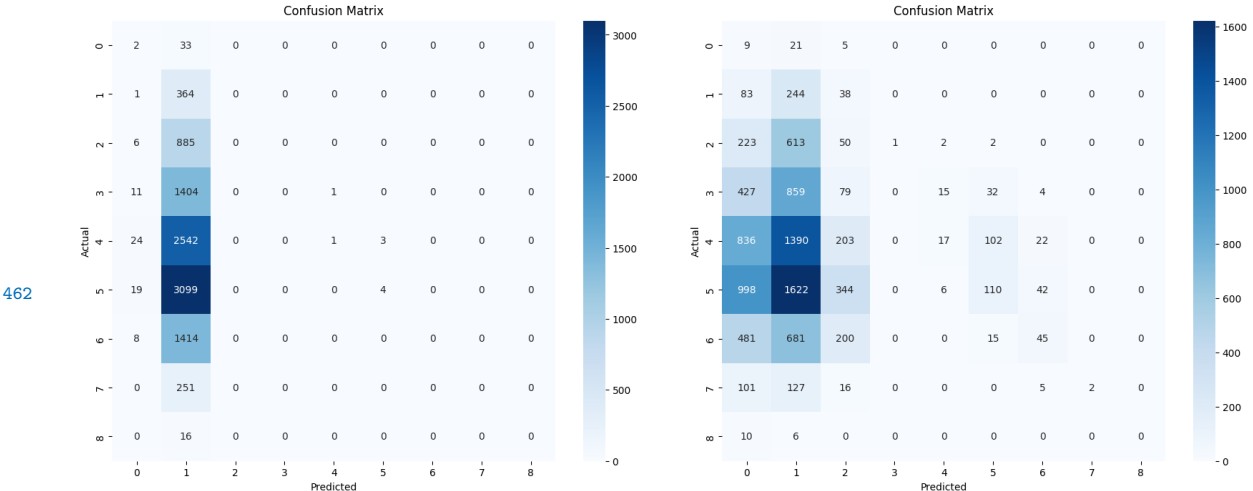

**Figure A.1.** Comparison of ResNets when Attacked by FGSM

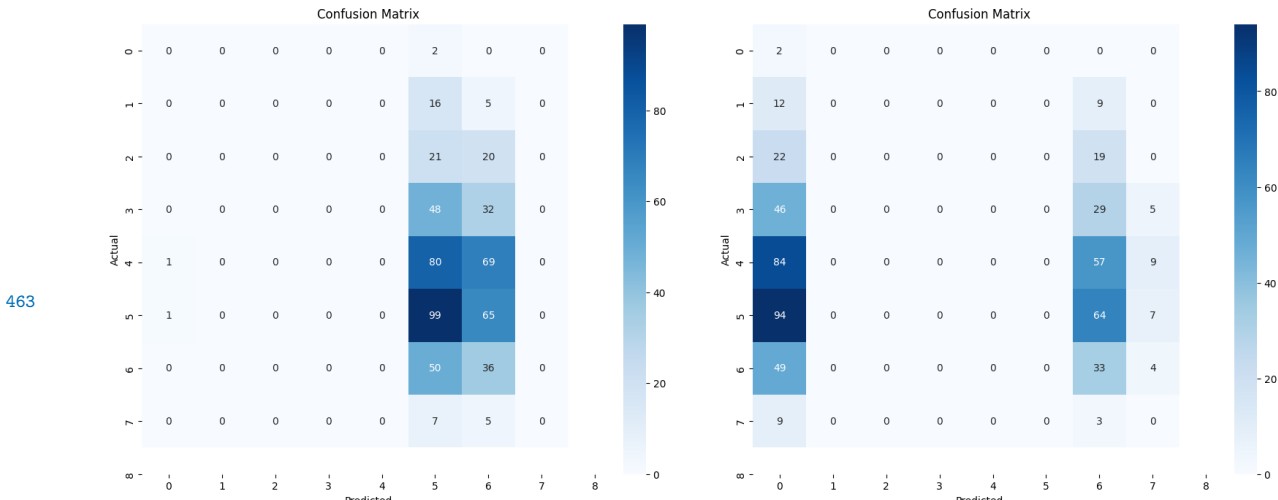

**Figure A.2.** Comparison of ResNets when Attacked by C&W

