# OpenReview forum: "Blessing or a Curse. Discussing Security Concerns of Diagnostic Models in Radiological Assessment"
_NLDL.org/2025/Conference — Submitted to NLDL 2025_

### Official Review · Reviewer_5fNc · 2024-10-09
**Review of Paper 10**

**Confidence:** 4

**Summary:**

This paper analyses the security risks of AI in radiological assessment. Specifically, the authors evaluate the risks of data poisoning attacks using Fast Gradient method and Carlini Wagner method on YOLO model. Further, attack detection using Benford's Law and mitigation via defensive distillation are also analysed.

**Strengths:**

* The paper targets an important field of research i.e analysing the risks of AI in healthcare diagnostics.
* The experimentation for attack detection as well as defense is good enough to understand the impact of both for the specific model (YOLO) and the dataset (RSNA Pediatric Bone Age Challenge) used in the work.

**Weaknesses:**

* The structuring as well as writing of the paper is ambiguous and difficult to follow. For example, the background section contains the methodology used in the work. Section 2.1 is basically a reference to the YOLO model, which is a methodology used in this work, instead of the background. Similarly, Section 2.2-2.5 also mention the methods used in this work but fail to do a literature review of related works.
* The motive behind using models such as YOLO as well FSGM and C&W are not described in the paper.
* There are several errors in the text, for example: 1) "the" written as "he" in line 22, 2) defining abbreviations multiple times, such as FGSM in line 311 and 367, 3) redefining C&W abbreviation as CW, 4) not referring to the figures when talking about them in the text (e.g line 253).
* Why are 500 and 1000 the number of adversarial examples generated for C&W and FSGM, respectively?
* As pointed by the authors, the YOLO model has only been trained for 10 epochs which might not be sufficient to evaluate the model against these attacks as well defenses.
* All experiments are based on only one dataset as well as model. More experimentation on a new dataset as well as model would lead to more conclusive results.

**Justification:**

The idea and methods used in the work are not well motivated by the authors as well as insufficiently experimented on only one dataset as well as model. The work also needs restructuring as well as rewriting.

---

> ### Author Rebuttal · Authors · 2024-10-25
>
> I thank the reviewer for their feedback. I added a second dataset to the analysis as per your recommendation. I also added literature review and attempted to target some of the reasoning for using YOLO in particular. Your review was very helpful, and I am really thankful that you made a bulleted list to help me target individual problems.

---

### Official Review · Reviewer_CVvV · 2024-10-09
**Discussion on vulnerability of YOLO model to data perturbations in radiology**

**Confidence:** 4

**Summary:**

The paper addresses the topic of vulnerability of radiology AI to malicious data perturbations. Overall, the paper is more of a discussion than a study, but it does present a case study with two different artificially generated perturbations for radiology images and the impact on YOLO detection performance, as well and a method for detecting the data fraud. While the study addresses a valuable research question, it falls short of providing thorough analysis or being useful for practical applications.

**Strengths:**

The topic is timely and worth addressing: robustness towards changes in the data domain, caused either by intrinsic shift or changes in sample characteristics, or by malicious manipulation, threat the trustworthiness of AI.

**Weaknesses:**

The current study reads more like a discussion rather than a thorough analysis of the topic. The presented cases with YOLO detector and presence of two kinds of manipulations is not very convincing. The experimental part should be significantly strengthened.

The article is not very well organized. It appears more useful as a discussion around the theme, but as such it is not well suited for this forum.

**Justification:**

Based on the experimental results, the paper does not provide very valuable information for any practical use. As a discussion, it is somewhat valuable, but strengthening the experimental part is recommended.

---

> ### Author Rebuttal · Authors · 2024-10-25
>
> I thank the reviewer for their feedback. I have restructured the article and added a second dataset to my analysis to strengthen the experimental aspects of the article.

---

### Official Review · Reviewer_484T · 2024-10-10
**Artificial Intelligence: A Blessing or a Curse. Discussing Security Concerns of Diagnostic Models in Radiological Assessment**

**Confidence:** 4

**Summary:**

This paper looks at the effect of poisoning the continuous training of an AI with data samples generated with two different techniques.

**Strengths:**

The problem deserves analysis and attention.

**Weaknesses:**

The results are anecdotal and no firm conclusions can be made
Continuous training of radiological AIs is far from the reality, as such models will not be regulatorical cleared
The paper is not written authoritatively
The two models of attack are simple
Only one data set has been used

**Justification:**

See weaknesses

---

> ### Author Rebuttal · Authors · 2024-10-25
>
> I thank the reviewer for their feedback. In my revisions, I attempted to expand the analysis with an additional dataset to strengthen conclusions.  I would like to respectfully argue that continuous training is not a far reality. The MONAI Deploy extension is starting to be used in clinical settings in the US making data attacks in healthcare a more serious problem.

---

### Meta-Review · Area_Chair_ZZWh · 2024-11-05

**Recommendation:** Reject
**Confidence:** 5

**Metareview:**

The submitted paper proposes to study several mitigation methods for data poisoning in the context of medical imaging.
The authors make the following claims in their introduction
- investigate adversarial attacks, focusing on FGSM and C&W attacks in the context of x-ray images classification and GI images segmentation
- better understand the applicability of Benford’s Law to medical imaging
- make use of distillation and study its impact on adversarial attacks

As pointed out by the reviewers, these claims are not backed by the experiments and analyses performed in the paper. The results regarding Benford's law are incomplete: the original digit distribution is missing, rendering conclusions about the usability of this method questionable. Results regarding distillation are inconclusive at best, probably due to the very short training. The choice of the methods studied is also questionable according to the reviewers as more sophisticated attacks exist.
In addition, the bibliography is missing from all revisions of the paper.
While the subject of the paper is of interest, there are too many structural and experimental issues for the paper to be accepted, I therefore recommend its rejection.

**Suggested Changes To The Recommendation:**

2: I'm certain of the recommendation.  It should not be changed

---

### Decision · Program_Chairs · 2024-11-06

Reject